# NTL-Unet: A Satellite-Based Approach for Non-Technical Loss Detection in Electricity Distribution Using Sentinel-2 Imagery and Machine Learning

**DOI:** 10.3390/s24154924

**Published:** 2024-07-30

**Authors:** Matheus Felipe Gremes, Renato Couto Gomes, Andressa Ullmann Duarte Heberle, Matheus Alan Bergmann, Luísa Treptow Ribeiro, Janice Adamski, Flávio Alves dos Santos, André Vinicius Rodrigues Moreira, Antonio Manoel Matta dos Santos Lameirão, Roberto Farias de Toledo, Antonio Oseas de C. Filho, Cid Marcos Gonçalves Andrade, Oswaldo Curty da Motta Lima

**Affiliations:** 1Department of Chemical Engineering, State University of Maringá (UEM), Maringá 87020-900, PR, Brazilocmlima@uem.br (O.C.d.M.L.); 2Pix Force Tecnologia S.A, Porto Alegre 90240-200, RS, Brazil; 3Light Serviços de Eletricidade S.A, Rio de Janeiro 20211-050, RJ, Brazil; 4Department of Electrical Engineering and Computer Science, Federal University of Piauí—UFPI, Teresina 64049-550, PI, Brazil; antoniooseas@ufpi.edu.br

**Keywords:** orbital monitoring system, non-technical losses (NTLs), electricity distribution networks, Sentinel-2 satellite imagery, computer vision, urban areas segmentation

## Abstract

This study introduces an orbital monitoring system designed to quantify non-technical losses (NTLs) within electricity distribution networks. Leveraging Sentinel-2 satellite imagery alongside advanced techniques in computer vision and machine learning, this system focuses on accurately segmenting urban areas, facilitating the removal of clouds, and utilizing OpenStreetMap masks for pre-annotation. Through testing on two datasets, the method attained a Jaccard index (IoU) of 0.9210 on the training set, derived from the region of France, and 0.88 on the test set, obtained from the region of Brazil, underscoring its efficacy and resilience. The precise segmentation of urban zones enables the identification of areas beyond the electric distribution company’s coverage, thereby highlighting potential irregularities with heightened reliability. This approach holds promise for mitigating NTL, particularly through its ability to pinpoint potential irregular areas.

## 1. Introduction

Measuring total losses in electricity distribution, resulting from the difference between energy injected into the distributor’s grid and energy actually supplied to regulated consumers, excluding energy allocated to free consumers, is an inherent challenge throughout the transformation, transport, and distribution process of this vital resource. When these losses reach significant levels, they impose additional demand, ultimately burdening regulated consumers for electricity generation. In general, the long-term marginal cost of generation is substantially higher than investments associated with reducing distribution losses [1]. Therefore, it is the responsibility of sector stakeholders to minimize these losses in order to optimize system efficiency and, consequently, provide benefits to society.

Non-technical losses (NTLs) in electricity distribution refer to a portion of total losses unrelated to technical factors during energy supply to consumer units. These losses are defined as the energy consumed by clients that are not billed by the utility [2]. They encompass issues such as theft, measurement inaccuracies, billing process issues, and the absence of metering equipment [3]. They can be categorized into three main components: commercial losses, related to measurement failures in regular consumer units; losses due to consumption from inaccessible clandestine connections, caused by unauthorized energy consumption without a formal contract, especially in restricted areas; and other technical losses, which are indirectly caused by NTL and are pragmatically considered part of the former [4,5].

NTL stemming from theft, fraud, reading errors, measurement inaccuracies, and billing discrepancies, are observed globally, especially in countries with low socioeconomic indicators [6,7]. According to a report by ANEEL [1], in 2020, the total amount of NTL across all Brazilian electric sector concessionaires amounted to BRL 8.4 billion. Several studies have documented the diverse impacts of NTL, with the primary concern being revenue loss, often resulting in utility companies transferring these costs to end consumers through tariffs [8,9]. Brazil’s vast expanse poses challenges for field surveillance. The electric distribution company Light serves approximately 4.5 million consumer units (CUs), indicating a high population density concentrated in urban areas, thus presenting a significant opportunity for optimizing field inspection team deployment.

These losses result in significant revenue reductions and adversely impact grid reliability [8,10]. NTL can overload transformers, thereby affecting the overall operation of the power system. Additionally, they compromise the quality of electricity supply, leading to issues such as voltage violations, infrastructure damage, blackout risks, and potential threats to public safety [8,9,10,11,12,13]. Consequently, reducing NTLs will also decrease the physical losses within the grid [14]. Furthermore, the costs associated with on-field inspections to recover these losses must be considered. The low effectiveness of these inspections can further escalate the costs related to NTL. Therefore, it is crucial for utilities to improve their strategies in this area and enhance the success rate of future on-field inspections to effectively recover revenue losses [2].

With recent technological advancements, modern techniques for detecting buildings and urban areas can lead to results with a high degree of accuracy. Rule-based methods utilize various properties of buildings, such as geometric, spectral, textural, contextual, and vertical characteristics, to identify them in images [15]. In contrast, image segmentation methods partition the scene into non-overlapping segments and identify buildings as objects of interest. These techniques can achieve satisfactory results with moderate manual input [16,17].

Building upon the aforementioned, this study proposes the development of an orbital monitoring system for urban area segmentation in Sentinel-2 satellite images, focusing specifically on regions of Brazil. By analyzing images and geospatial data, we employ computer vision techniques and machine learning algorithms to achieve this objective. The automated delineation of urban areas enables energy companies to prioritize surveillance efforts in regions with a higher risk of theft, thereby optimizing resource allocation and enhancing efficiency in combating energy theft.

The concession area considered in the research exhibits significant economic heterogeneity and encompasses high-risk regions, which complicates the provision, maintenance, and operation of the electricity supply service by operational teams. These regions are designated as areas with severe operating restrictions (ASROs) [18].

The proposed method has applicability in various fields of study, such as urban planning, deforestation monitoring, and monitoring the growth of urban areas, which is the focus of this work. Therefore, we highlight the following contributions:An automatic computational method for identifying irregularities in urban areas; andA method based on public data, capable of directing the efforts of energy companies to combat NTL, by identifying possible irregular areas, including ASROs.

The subsequent structure of this work is outlined as follows: In Section 1, we contextualize the present study in relation to related works. Section 2 details the developed methodology, while Section 3 and Section 4 present and discuss both quantitative and qualitative results. We conclude the work in Section 5 with conclusions and suggestions for possible future work. Significantly, the method shows promise, demonstrating efficient performance in new regions and successfully generalizing to unexplored areas. It effectively identifies urban regions and, consequently, images beyond the coverage area of the electric distribution company, indicating potential irregularities.

### Related Works

To provide a comprehensive overview of the evolution of methods developed for this purpose, we will present related works to the topic addressed in this research. Land cover classification is commonly addressed as a task of pixel categorization in the remote sensing community [19].

In the method developed by [20], the authors employed a hybrid Convolutional Neural Network (CNN) model to classify land cover using Landsat 8 images, achieving an overall accuracy of 92.5%. Similarly, ref. [21] utilized a U-Net architecture to classify high-resolution remote sensing data, achieving an overall accuracy of 94.6%. In a separate investigation, ref. [22] utilized a methodology based on Deep Learning (DL) to conduct land cover classification using Sentinel-2 satellite data, resulting in an overall accuracy of 97.8%.

In the study proposed by [23], a multitask-learning framework for classifying urban land cover was proposed, employing high-resolution images, achieving an overall accuracy of 91.3%. Meanwhile, in the study proposed by [24], a DL-based method is utilized to detect changes in land cover from multi-temporal Landsat 8 data, achieving an overall accuracy of 96.7%.

Traditional machine learning classifiers such as decision tree (DT), support vector machine (SVM), and random forest (RF) have been employed for land cover classification [25]. Comparative studies have highlighted the superiority of DL-based approaches over traditional machine learning methods in terms of accuracy [22,25], achieving the highest overall accuracy of 97.8% for Sentinel-2 data. These approaches encompass a variety of DL architectures, including CNNs, deep belief networks, and attention-based models.

Land cover analysis is commonly conducted through remote sensing, where data such as satellite images play a central role. This approach enables monitoring changes in land cover, but the accuracy of the results is subject to variables such as the type of sensor used, image resolution, and classification algorithm employed.

The implementation of DL techniques represents a remarkable advancement in the effectiveness and quality of region monitoring, providing highly accurate classifications across various datasets and scenarios. These innovative approaches stand out for their contribution to enhancing environmental monitoring, resource management, and disaster prevention. However, it is important to note that these methods require substantial computational resources and comprehensive training datasets. In this context, to mitigate the challenge of large databases, the present research focuses on investigating the U-Net, exploring various synthetic data augmentation operations and diverse encoders. Thus, optimizing hardware resources with the combination of hyperparameters that results in a robust and versatile method.

Our approach introduces an orbital monitoring system that utilizes Sentinel-2 satellite imagery for the automated segmentation of urban areas, specifically tailored to regions in Brazil. By integrating computer vision techniques and machine learning algorithms, we aim to enhance surveillance capabilities and prioritize resource allocation in areas prone to NTL. This methodology facilitates the identification of irregularities, such as energy theft and unauthorized connections, and also optimizes operational efficiencies for electric distribution companies. Thus, our study aims to contribute to the mitigation of NTL, thereby improving overall grid reliability and reducing financial burdens on regulated consumers.

## 2. Materials and Methods

Our methodology comprises four main steps: (i) data acquisition from Sentinel-2 images available in the Copernicus repository [26]; (ii) training of a DL architecture, specifically the U-Net; (iii) construction of a land cover area using public data provided on the distributor’s geographic database (BDGD) portal [27]; and, finally, (iv) evaluation of the results obtained. Figure 1 illustrates the workflow followed.

### 2.1. Image Acquisition

The dataset used for model training was the MultiSenGE [28]. Among the datasets found in the literature, MultiSenGE stands out as the only semantic segmentation dataset utilizing Sentinel-2 images with high precision and quality annotations [29], in addition to featuring a large number of images for training. Although the dataset pertains to the region of France, it is expected that the model can extrapolate the results so that urban areas are correctly segmented. Another prominent factor in using MultiSenGE corresponds to seasonal changes, since this dataset includes images from all four seasons, ensuring that the model can accurately segment urban areas, regardless of seasonal variability. Finally, it is worth noting that no public dataset meeting the described characteristics was found for regions of Brazil.

#### 2.1.1. Test Dataset

To deepen the analyses of the proposed method, we chose to focus on a specific region, selecting the city of Rio de Janeiro. This region offers rich detail, enabling comprehensive validation of our methodology through a wide range of tests.

Comparing the regions of Rio de Janeiro and the area of France covered by the MultiSenGE dataset visually reveals significant differences in geographical characteristics. Rio de Janeiro’s urban landscape is characterized by its rugged mountainous terrain, resulting in irregular urban layouts and varying elevation levels. In contrast, the area of France covered by the MultiSenGE dataset typically exhibits flatter or gently rolling terrain, fostering a more uniform urban structure and street layout.

Utilizing a Sentinel-2 image captured on 22 July 2023, we observed coverage of various regions, including the city of Rio de Janeiro, within a single tile (the specific spatial unit of image data captured by the Sentinel-2 satellite), as depicted in Figure 2. Upon analysis, we identified an extensive maritime area, prompting us to exclude it from consideration. The ROI was empirically defined to encompass a relevant built-up area, as depicted in Figure 2. Covering an area of 4000 km², the ROI’s coordinates [629,869.0, 7,446,400.0, 709,178.0, 7,491,689.0] were delimited using the spatial reference system CRS: EPSG:32723—WGS 84/UTM zone 23S.

The land cover masks were generated based on OpenStreetMap (OSM) annotations [30], which serve as the Ground Truth for the test image set. By leveraging specific OSM tags such as “residential”, “industrial”, “commercial”, “airport”, “road”, and “building”, we identified corresponding regions for these categories. These tags offered a comprehensive and precise representation of various land cover types, facilitating the accurate creation of masks for analysis. Following the automated generation of preliminary annotations, a meticulous manual correction process was conducted to enhance annotation accuracy and quality. The process involved two human annotators with expertise in the field, who manually corrected the annotations using specific tools assisted by Sentinel-2 images. Figure 3 visually demonstrates this process, showcasing the Sentinel-2 Tile, preliminary annotations highlighted in yellow, and manual corrections depicted in red and blue. The red markings represent regions where the yellow annotations were incorrectly marked, while the blue markings indicate areas missed by the yellow annotations.

The final outcome is a comprehensive and accurate dataset representing urban areas in the Rio de Janeiro region. Figure 4 illustrates an example of the image mask generated from urban annotations. To construct it, an image acquired on 22 July 2023 was utilized. This dataset comprises 2232 256 × 256-pixel chips, each containing information for the 12 Sentinel-2 bands with a spatial resolution of 10 m.

#### 2.1.2. Image Preprocessing

For each image from the Sentinel-2 mission, both the training and testing sets possess the following properties:Different spatial resolutions: These resolutions are categorized into three levels: 60 m, 20 m, and 10 m.Images for 12 bands: Each captures specific information about the Earth’s surface. These band images include multispectral data ranging from near-infrared to ultraviolet, providing a wide range of information for detailed analyses.Information about regions with clouds and/or similar features such as ice, through the Sent2Cor tool [31].

Accordingly, before initiating the model training process proposed in this study, we applied a reprojection procedure to the images. This process aims to standardize the resolution of all bands to 10 m, ensuring consistency by eliminating areas that may negatively impact the final result. Finally, as part of pre-processing, we remove regions with cloud cover using the information contained in the Sentinel-2 scene classification map (SCL) file. SCL is a derivative product that classifies each image pixel into one of 11 classes, including the cloud class. This process allows for the automatic identification and exclusion of areas affected by clouds, ensuring that only high-quality data is used in semantic segmentation. For greater accuracy, we used a confidence threshold to ensure that partially cloudy or cloud-shadowed pixels were also removed [31].

#### 2.1.3. Impact of Spatial Resolution

This study utilizes Sentinel-2 satellite imagery with a 10-m spatial resolution, which is suitable for urban area detection. However, smaller built-up areas could be more effectively detected with higher-resolution imagery. Using satellite images with resolutions finer than 10 m would likely enhance the detection of isolated constructions, as these smaller structures would be more discernible in higher-detail images. Consequently, the proposed method’s accuracy could improve with such data.

Despite these potential benefits, we chose to use Sentinel-2 imagery primarily because it is freely available. In contrast, higher-resolution satellites often incur significant costs. Additionally, Sentinel-2 offers a favorable revisit time, capturing images of the same location every 5 days. This frequent data acquisition is crucial for ongoing monitoring and timely updates, making Sentinel-2 a practical choice for our study’s objectives.

### 2.2. Urban Area Segmentation

The architecture of CNN U-Net was proposed by Ronneberger et al. [32] for semantic segmentation purposes. The structure of the U-Net is characterized by an inverted “U” shape, consisting of an encoder and a decoder [32]. The original construction of this network consists of 23 convolutional layers interleaved by ReLU and maximum pooling operations. The architecture obtains its name because it was conceived in a structure that resembles the letter U. The descent is a path of contraction, and the ascent is a path of expansion.

The encoder, responsible for feature extraction, consists of convolutional layers followed by pooling operations for dimensionality reduction. The equations associated with this stage involve convolutions (Conv) and pooling operations (Pool), as depicted in Equation (Equation 1).
(1)features=Pool(Conv(Input))

The decoder, responsible for reconstructing the segmented image, consists of upsampling layers followed by convolution operations. The equations include the upsampling (UpSample) and convolution (Conv) operations, as illustrated in Equation (Equation 2).
(2)segmentation=Conv(UpSample)

The loss function commonly used in U-Net training is generally the cross-entropy (CE), which measures the discrepancy between the network’s prediction and the true segmentation mask as depicted in Equation (Equation 3).
(3)$$CE\left(y_{\mathrm{pred}},y_{\mathrm{true}}\right)=-\sum y_{\mathrm{true}}\log\left(y_{\mathrm{pred}}\right)$$

In summary, the U-Net achieves effective image segmentation by learning to map inputs to corresponding segmentation masks through the combination of encoding and decoding layers, utilizing ReLU activation functions, and minimizing cross-entropy as the loss function during training.

Widely employed in various image segmentation tasks, the U-Net has been utilized for segmenting cells in medical images, objects in satellite images, and structures in microscopic images, among others. Its ability to precisely segment images with high efficiency renders it a powerful tool in a broad range of applications where object segmentation is crucial [32]. Furthermore, the U-Net demonstrates high adaptability, as it can be constructed using established convolutional neural network backbones such as ResNet, EfficientNet, Vision Transformer, and others [32,33].

#### Network Training

During the training phase of the U-Net architecture, a meticulous approach was adopted to optimize the performance of the neural networks. A wide range of parameters was experimented with to enhance the model’s effectiveness. Among the key hyperparameters adjusted, the following are highlighted:Learning rate (LR):−$1\times 10^{-4}$.−$1\times 10^{-3}$.Loss function (LF)):−Dice loss.−Cross-entropy loss with and without weights.−BCE With logit loss.

Other hyperparameters, such as batch size and input image dimensions, were adjusted based on data availability and hardware resources. The specific configurations of these hyperparameters may vary depending on the nature of the dataset and the capabilities of the system used for training. For the problem at hand, images with dimensions of 256×256 were chosen, as provided in the dataset.

Additionally, data augmentation operations were applied to diversify and increase the data samples, aiming to enhance the model’s ability to generalize more robustly. These augmentations were exclusively applied to the training set and can be categorized into two classes: spatial-level transformations and pixel-level transformations.

Spatial-level transformations:−Vertical and horizontal flipping.−Random cropping.Pixel-level transformations:−Blurring.−Channel shuffling.−Random brightness and contrast adjustment.

As described in Section 2.1, each study image comprises 12 bands containing information from various spectra. In this context, several band combinations were tested to determine the one with the best performance in the task of detecting built-up areas. Additionally, we varied the U-Net backbones with different encoders, along with their respective variants. The following combinations stand out.

Specific Band Combinations:−RGB.−NIR (near-infrared).−SWIR (shortwave infrared).Encoders:−EfficientNet.−MixVisionTransformer.−ResNet.

All experiments involved the use of pre-trained weights provided by ImageNet [34]. This approach aims to find the optimal band configuration and the most suitable encoder architecture for efficient detection of built-up areas, providing a comprehensive analysis of the different combinations and architectures used in the experiments.

Overall, choosing hyperparameters is not a trivial task, and our method tested various possible combinations to mitigate data effects. Below, we outline the main motivations behind the hyperparameter choices:Learning rate (LR): We opted to test two learning rates: $1\times 10^{-4}$ and $1\times 10^{-3}$. The $1\times 10^{-4}$ rate was chosen to ensure fine and stable adjustment of the network weights, while $1\times 10^{-3}$ was tested to accelerate initial convergence, allowing rapid exploration of promising parameter spaces.Loss function (LF): We tested three loss functions that best adapt to the segmentation problem, each with its specificities:−Dice loss: Handles imbalanced classes well by maximizing intersection over union of predictions and ground truth.−Cross-entropy loss (with and without weights): Evaluates pixel classification effectiveness, with the weighted version helping to mitigate the impact of imbalanced classes.−BCE With logit loss: Effective in binary classification problems, with embedded sigmoid function for numerical stability.Batch size and input image dimensions: We opted for 256×256 images as provided in the dataset, to balance adequate resolution for precise detection and computational efficiency. The batch size was adjusted based on data availability and hardware resources.Data augmentation: We explored combinations of different techniques, both for spatial and pixel-level operations, to increase the diversity and quantity of data samples.Band combinations: We investigated isolated band combinations that yielded the best results for the technique used, utilizing all available bands;Encoders: We tested encoders with different properties to investigate the best technique for the problem at hand. Choosing EfficientNet, ResNet, and MixVisionTransformer was due to their successful performance in other image segmentation tasks.Finally, we used pre-trained weights provided by ImageNet to initialize our models, leveraging previously acquired knowledge to enhance the efficiency and performance of initial training.

### 2.3. Coverage Area

This section outlines the process by which the service area covered by a particular energy concessionaire is measured. The coverage area was determined using the geographic database of the distributor (BDGD), an essential component of the Regulatory Geographic Information System for energy distributors in Brazil (SIG-R) [27]. Cartographic data are provided in a geodatabase format (.gdb) per distributor, facilitating access and management of geographic information. The BDGD Instruction Manual and Module 10 of the PRODIST offer detailed guidance on the BDGD structure and the appropriate utilization of its entities and fields [35].

In this project, geospatial data from power lines is utilized to delineate the coverage area of the electrical network. The representation of the potentially covered area is achieved through the establishment of a buffer zone encircling the transmission and distribution lines. This buffer encompasses an additional spatial extent around the lines, serving as a safety margin to assess and address potential impacts or associated activities. The primary objective of this method is to identify geographical regions distanced from electrical infrastructure, where occurrences of invasions or energy theft, known as NTL, are more likely. The dimensions of the buffer zone surrounding the transmission and distribution lines significantly influence the extent of the potentially covered area. A larger buffer leads to a broader coverage area, thereby reducing the number of identified areas susceptible to invasion.

In Figure 5, the electrical coverage area with a 100-meter buffer applied to the low- and medium-voltage lines of the BDGD is depicted. The areas covered by the lines are highlighted in green, providing a clear visualization of the extent of electrical coverage in the study area.

#### Integration of Urban Area Segmentation and BDGD

After segmenting the urban areas, we utilized BDGD data to generate a polygon delineating the energy concessionaire’s coverage area. Next, we applied the urban segmentation mask to this polygon, subtracting the segmented areas from the concessionaire’s coverage. This subtraction unveils intrusion regions, with potential for irregular occupation and NTL, enabling effective identification of suspicious areas to direct monitoring and regularization efforts. It is important to note that the indicator of areas suggestive of invasion is based solely on the subtraction of these polygons, and no other indicator is taken into account.

Figure 6 visually illustrates this process by depicting the overlay of the concessionaire’s coverage polygon (highlighted by the areas outlined in green) with the identified urban area mask highlighted in yellow.

### 2.4. Evaluation of Results

Evaluating semantic segmentation models plays a crucial role in analyzing and comparing different neural network architectures for image segmentation tasks. These metrics enable the assessment of the model’s performance and accuracy in identifying and delineating objects of interest in an image. Among the most commonly used metrics for this purpose are accuracy (Acc), precision, recall, F1 score, and intersection over Union (IoU) [36].

In a comprehensive manner, we present an automated method for detecting urban areas that may exceed the coverage designated by the concessionaire, thus requiring further analysis.

## 3. Results

We will provide a detailed presentation of the results obtained through the application of the proposed methodology (Section 2). All experiments were conducted using hardware with the following specifications: 62 GB RAM, NVIDIA RTX A5000 24 GB, and Intel(R) Xeon(R) Gold 6240R CPU 2.40 GHz

### 3.1. Definition of the Base Model

To initiate the training of the U-Net, numerous experiments were undertaken to establish the initial configuration and optimize hardware utilization. For model training, we only used the following classes: dense built-up (1), sparse built-up (2), specialized built-up areas (3), specialized but vegetative areas (4), and large-scale networks (5), as they contain the necessary information for the model to specialize in detecting urban areas. These subclasses were combined to form a single class (urban area). It is worth noting that our segmentation network has only two classes: “urban area” and “non-urban area”. Additionally, images from the MultiSenGE dataset were randomly split into 70% for training, 15% for testing, and 15% for validation.

The strategy initially involved defining the basic hyperparameters. In this regard, we attempted to determine the learning rate (LR) and the loss function (LF) that exhibited the best performance in our testing scenario.

Learning rate (LR):−$1\times 10^{-4}$.−$1\times 10^{-3}$.Loss function (LF):−Dice loss.−Cross-entropy loss with and without weights.−BCE With logit loss.

Due to the large number of possible experiments, a batch size (BS) of 64 was set. Another factor influencing processing time directly is the number of epochs. Therefore, we chose to run 50 epochs per experiment. We determined the optimal number of epochs through prior experimentation. The optimizer used was Adamax, with a scheduler with a weight decay set at 0.001 and a patience of 15 epochs. The bands used were corresponding to red, blue, and green (RGB). Finally, initial augmentations were applied, such as vertical and horizontal flips.

Table 1 presents a summary of the metrics achieved in this testing phase, which aided in defining the hyperparameters for subsequent experiments.

As shown in Table 1, based on the IoU metric, the best result was achieved for the cross-entropy loss function, with an LR of $1\times 10^{-4}$. Figure 7 presents random results obtained for the initial batch of the test set.

As depicted in Figure 7, several errors are noticeable, highlighting the necessity for improved outcomes. Specifically, while the segmentation network effectively delineates the urban area located closer to the center, it demonstrates limitations in accurately detecting the more peripheral urban regions. This discrepancy suggests a need for refinement in the segmentation model. Initially, we considered all subcategories of “Urban Areas”, encompassing “Dense Built-Up”, “Sparse Built-Up”, “Specialized Built-Up Areas”, “Specialized but Vegetative Areas”, and “Large-Scale Networks”. However, through a thorough examination and numerous tests, we recognized the potential for more promising results by concentrating solely on the categories of “Dense Built-Up”, “Sparse Built-Up”, and “Specialized Built-Up Areas”, which denote significant urban areas. This determination stemmed from empirical observations and previous experiments.

We repeated the experiment using only these three categories, as mentioned in [28]. Among the top-performing results highlighted in Table 1, employing the cross-entropy loss function and a learning rate of $1\times 10^{-4}$ exhibited the best performance. This led to improvements in the numerical outcomes, as demonstrated in Table 2. Figure 8 presents some visual results for the experiment described.

After defining the best hyperparameters (LR: $1\times 10^{-4}$; LF: CrossEntropyLoss; using the standard ResNet18 encoder), further experiments were conducted. We detail each of them as follows:Augmentations using only spatial-level transforms.−Vertical flip (already used).−Horizontal flip (already used).−Random cropping.Augmentations using only pixel-level transforms.−Blurring.−Channel shuffling.−Random brightness and contrast adjustment.All augmentation operations combined.

Table 3 presents the results of this set of experiments.

Table 3 reveals that the outcomes from alternative data augmentation approaches did not yield significant positive effects, as evidenced by the absence of improvement in the results. Consequently, we proceeded with training using exclusively spatial augmentation operations. Among the considerations, we emphasize the inadequacy of the chosen augmentation techniques for the specific problem context, the potential introduction of noise or undesired distortions in the images during the augmentation process, or the mismatch between the selected techniques and the problem’s inherent nature. These factors may hinder the augmentation operations from enhancing the model’s generalization capacity, thereby resulting in a lack of improvements in the outcomes.

After presenting the outline earlier, we conducted additional experiments involving various combinations of bands. These experiments were conducted considering the following band variations:Experiments considering band variations:−All bands.−RGB, NIR, and SWIR.

Table 4 presents the results of these band combinations.

Based on the results presented in Table 4, we note that, up to this point, there have been no significant performance improvements compared to previous experiments. Specifically, the IoU metric considered the most relevant indicator, did not show a significant improvement compared to the best result previously achieved. Given this finding, we have decided to maintain the experimental settings that provided the best performance so far: a learning rate (LR) of $1\times 10^{-4}$ and a loss function (LF) of cross-entropy loss, as illustrated in Figure 8.

The remaining experiments continue using the U-Net with different encoders (ResNet, EfficientNet, and Mix-vision-transform). In Table 5, we present the corresponding values.

Finally, after analyzing the experiments detailed in Table 5, where we evaluated the performance of the U-Net combined with various encoders, we chose to select the combination that yielded the best results for each one, bearing in mind that the ResNet18 encoder has already been tested in the initial experiments, as it is the default encoder:-ResNet18-ResNet34-Mix-vision-transform-b3

#### Statistical Tests

To validate the choice of model hyperparameters, we opted to perform two statistical tests. First, a one-way ANOVA [37] (Table 6) to compare the different loss functions, i.e., Dice, CE, BCE, and CE wgt for a fixed LR of $1\times 10^{-4}$. This test will determine if there are significant differences in performance metrics (Acc, F1, IoU, and ) between the loss functions. Next, we performed an independent samples *t*-test [38] (Table 7) to compare learning rates of $1\times 10^{-4}$ and $1\times 10^{-3}$, using the CE loss function, to verify if a specific learning rate provides better performance. We chose to perform these two tests because they are fundamental choices that were used throughout the methodology and were established at the beginning of the experiment.

The results presented in Table 6 show significant differences in the performance metrics Acc, F1, IoU, precision, and recall among the different loss functions when the learning rate is fixed at $1\times 10^{-4}$. To account for multiple comparisons, the Bonferroni correction [39] was applied. All adjusted *p*-values remain below the standard significance level of 0.05, confirming the statistical significance of these differences.

The *t*-test results (Table 7) indicate significant differences in the performance metrics Acc, F1, IoU, precision, and recall between the two learning rates of $1\times 10^{-4}$ and $1\times 10^{-3}$, using the CE loss function. For Acc, precision, and recall, the *p*-values are lower than the standard significance level of 0.05, indicating that the differences are statistically significant, with a learning rate of $1\times 10^{-4}$ showing superior performance. For F1 and IoU, the *p*-values are slightly higher than 0.05, suggesting that the differences are nearly significant, still indicating a trend of better performance for the $1\times 10^{-4}$ learning rate.

In conclusion, both tests affirm the superior performance of using the CE loss function with LR of $1\times 10^{-4}$ across a range of evaluation metrics.

### 3.2. Results of the U-Net Applied to the Rio de Janeiro Region

After defining the necessary hyperparameters for model construction, we evaluated them in the ROI defined in Section 2.1.1 (Figure 2), namely, for the Rio de Janeiro region. Specifically, a subset of images corresponding to 2232 images from 22 July 2023, was chosen due to the absence of cloud cover in the ROI. Table 8 presents the results of the different combinations of encoders considered.

When applying the U-Net trained on the MultiSenGE dataset to the region of Rio de Janeiro, we observe promising results, with IoU scores exceeding 88% for the test image set. It is noteworthy that the U-Net was originally trained using data from France (MultiSenGE) and has now been applied to a different geographical area (Rio de Janeiro, Brazil). This scenario facilitates the evaluation of the model’s generalization capacity across diverse geographic contexts.

In Figure 9, we display the outcome of the network for the ROI. As can be seen, the numbers presented in Table 8 corroborate the qualitative outcome of the proposed methodology.

### 3.3. Results of Urban Area Segmentation versus BDGD

In this section, we investigate the relationship between urban area segmentation data and coverage information provided by the electric distribution company in the BDGD repository. Initially, we present the coverage area of the electric distribution company based on BDGD data for the ROI with a 100 m buffer.

In Figure 10, it is possible to observe the area segmented by the model proposed in the methodology advocated here (in red), compared to the coverage area of the electric distribution company (in yellow).

To identify probable invasion areas, we conducted a subtraction between the region formed by the union of the medium and low voltage network coverages and the result of the proposed methodology. This procedure reveals the ‘excess’ area that requires further investigation by the electric distribution company. In Figure 11, we can see the result for the entire region of interest.

As seen in Figure 11, many red areas lie outside the coverage area provided by the electric distribution company; these regions are what our methodology characterizes as potential invasion areas.

### 3.4. Case Studies

To facilitate the understanding of the results and investigate potential areas suggestive of invasion, we conducted detailed case studies. In these case studies, a buffer zone of 100 m around the transmission and distribution lines was selected to represent the potentially covered area. There are clandestine connections within the buffer zone with a distance of less than 100 m to the power grid. Therefore, this method is more suitable for critical regions such as Rio de Janeiro. This choice ensures that only areas considerably distant from the coverage area are identified as potential invasion zones. Given their distance from the coverage area, these regions are more suspicious of being potential invasion areas. However, it is important to highlight that the chosen buffer value directly impacts the number of regions suggestive of irregularities. The smaller the buffer, the greater the number of identified regions. Therefore, this parameter can be adjusted according to the needs and characteristics of the location (areas with higher or lower urban concentration) of the analysis to be conducted.

In Figure 12, we present an example of areas suggestive of invasion. The central point (approximate) is located at −22.845202, −43.531593, and the average area is 496,922.767 $m^{2}$ or $0.497\;{\mathrm{km}}^{2}$.

It is important to highlight that only the larger region (bounded by dashed lines) was considered for area calculation, but as can be seen, other regions also exhibit similar characteristics and require further analysis by the electric distribution company.

Continuing with the case studies, we now delve into the situation illustrated in Figure 13. Upon analyzing the example, it becomes evident that the region in question exhibits highly suggestive characteristics of invasion, as its growth extends toward the peripheral area.

Lastly, Figure 14 illustrates several areas exhibiting behavior suggestive of invasion.

## 4. Discussion

Based on the experimentation conducted in Section 3.1, Section 3.2 and Section 3.3, we observe that the method demonstrates promising efficacy. We will highlight the key points that we consider fundamental to the proposed methodology:The method employs images with a spatial resolution of 10 m per pixel, a crucial choice in the context of remote sensing. This resolution provides the essential capability to examine significant details in the urban environment.The proposed method exhibited significant performance on the test set, achieving remarkable results with IoU scores surpassing 88% in the task of urban area segmentation.Given the robustness and extensive exploration of the method, it is noteworthy that the U-Net’s approach allowed for the incorporation of various encoders. This facilitated the recognition of diverse areas, leading to the development of robust and versatile architecture.We believe that the Vision Transformer-based encoder produces superior outcomes owing to its capability to partition images into smaller “patches”, thereby enhancing precision in pixel classification during semantic segmentation.In addition to achieving promising results as previously presented, it is worth highlighting that the method is generic, enabling training in one region of the world and testing in another, providing positive indications of its generalization capability. This was evidenced by making predictions in the Rio de Janeiro region without the need for model retraining.The strategy used to construct the coverage area with BDGD data was crucial, enabling the coverage of different electric distribution companies and the examination of various regions in Brazil.The results of areas outside the electric distribution company’s coverage demonstrate that the method efficiently indicates regions with limited urban area information.By exploring the case studies, we observe suggestive indications that the areas identified are indeed invasion areas.The precise segmentation of urban areas and the identification of irregularities can inform early warning systems, highlighting construction regions in risky or unauthorized areas or even possible invasions, which irregularly burden energy distribution networks, in addition to causing real risks to the lives of everyone involved.It is also possible to encourage detailed urban mapping that can assist in land use planning, ensuring that energy distribution networks are resilient and adaptable to future growth and environmental and urban challenges.Finally, the integration of this knowledge increases the reliability and safety of energy infrastructure, contributing to the effective management of geological risks.

The method demonstrates satisfactory performance in the presented case study, effectively identifying suspicious regions as potential areas of encroachment. This facilitates a more confident decision-making process by the electric distribution company.

In terms of potential improvements, it is important to note that some aspects need to be enhanced with this method. Figure 15 exemplifies certain regions that were identified as false positives.

Among the areas delineated in Figure 15 and considered as false positives, we have the following:Airport.Military areas.Public areas.

## 5. Conclusions

The proposed method demonstrated promising results in urban area detection, both in the training and test datasets. The effectiveness was evidenced by quantitative results, with IoU scores exceeding 88% in the task of urban area segmentation, and by qualitative results from the presented case studies.

The cloud removal strategy using masks available in the SCL allowed for a more thorough analysis, facilitating a more precise and reliable assessment of urban areas. This approach ensured more consistent and informative results.

In the context of constructing the test dataset, the use of masks available in OpenStreetMap as a form of pre-annotation proved to be significant. This allowed for the demarcation of areas independent of location, although some corrections are necessary due to the collaborative nature of the annotations in OpenStreetMap.

Apart from the promising segmentation results, the method proved to be generic, allowing training in one region of the world and successful application in another without the need for retraining. This generalization potential was confirmed through predictions in the Rio de Janeiro region.

The integration of different encoders in the U-Net approach, especially the use of a Vision Transformer-based encoder, enhanced precision in pixel classification during semantic segmentation, thanks to its ability to divide images into small “patches”.

The strategy employed for constructing the coverage area with BDGD data was crucial, covering different electric distribution companies and allowing the analysis of various regions in Brazil. The results obtained outside the coverage area of the electric distribution companies demonstrated that the method efficiently identifies regions with limited urban information, indicating possible irregularities. The case studies suggest that the identified areas are indeed invasion areas, which can have significant implications for the assessment and management of geographic risks.

Finally, the integration of this knowledge can increase the reliability and safety of energy infrastructure, contributing to the effective management of geological risks. Accurate identification of urban areas and irregularities is essential for land use planning and to ensure that energy distribution networks are resilient and adaptable to future challenges.

For future research aimed at enhancement, we propose:Model refinement to mitigate false positives, by considering specific characteristics of airports, military areas, and public spaces.Exploration of advanced image processing techniques to improve segmentation accuracy, particularly in challenging areas.Investigation of alternative indicators or data sources to complement the identification of areas susceptible to invasions or NTL. This could include analyzing factors such as energy consumption patterns, payment behavior, historical outage data, population density, economic activity, community reports, and the urban growth rate of the region.Assessment of the urban growth rate using the proposed methodology applied to past years’ data, providing insights into the expansion of urban areas and potential correlations with irregularities in energy consumption or distribution.Utilization of machine learning techniques to predict NTL hotspots, enhancing the precision of identifying high-risk areas.Integration of additional datasets to improve the understanding of urban energy consumption patterns, potentially including socio-economic data, climate data, and detailed infrastructure maps.

## Figures and Tables

**Figure 1 sensors-24-04924-f001:**
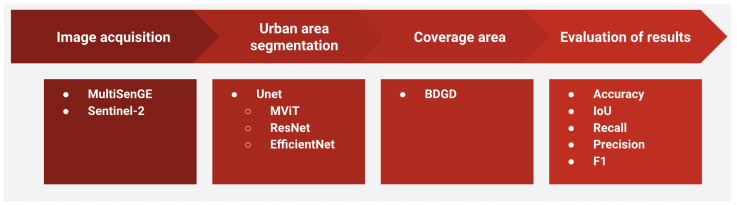
Proposed method.

**Figure 2 sensors-24-04924-f002:**
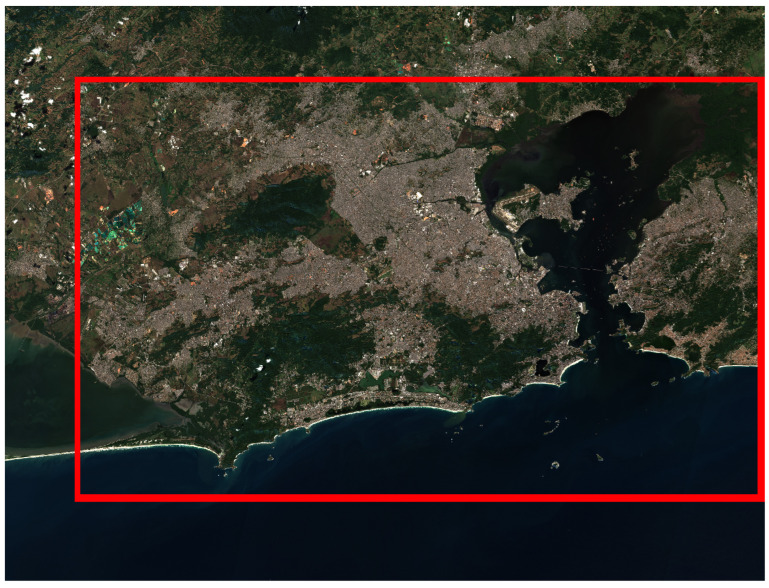
Delimitation of the region of interest (ROI) in the Metropolitan Area of Rio de Janeiro.

**Figure 3 sensors-24-04924-f003:**
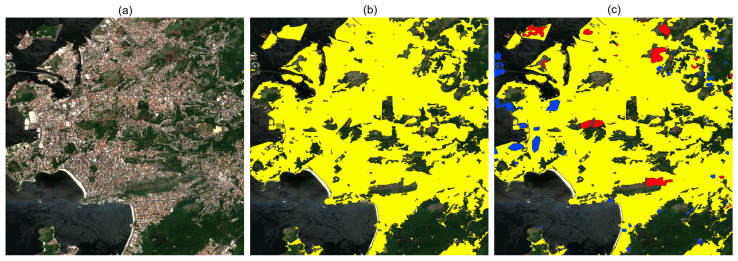
Result of the annotation process with OSM along with manual corrections. (**a**) RGB image of a region in Rio de Janeiro. (**b**) RGB image with OpenStreetMap pre-annotations shown in yellow. (**c**) The RGB image with human-made corrections is shown in red and blue.

**Figure 4 sensors-24-04924-f004:**
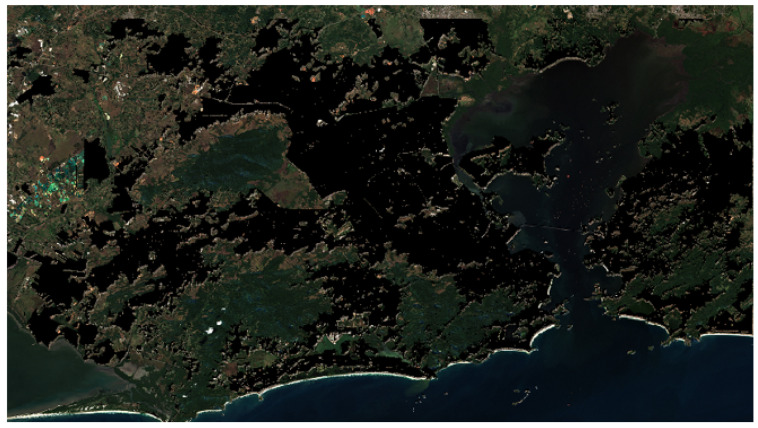
Rio de Janeiro region with annotated and corrected urban areas (urban area in black).

**Figure 5 sensors-24-04924-f005:**
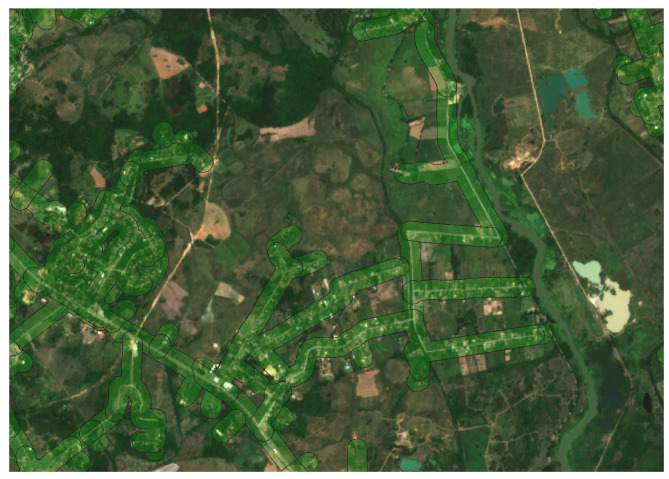
Electrical coverage area with a 100-meter buffer on the low- and medium-voltage lines of the BDGD.

**Figure 6 sensors-24-04924-f006:**
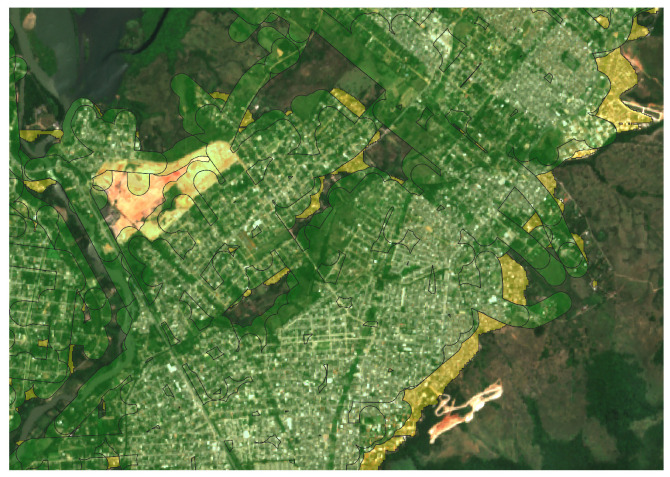
Coverage Polygon (in green) overlaid with Urban Area Mask (in yellow).

**Figure 7 sensors-24-04924-f007:**
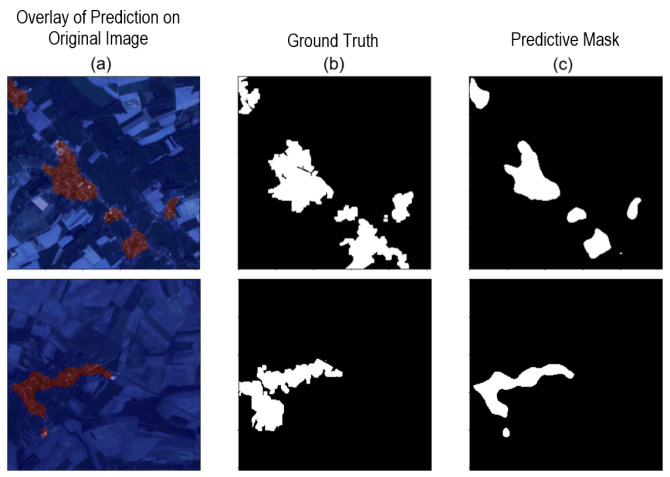
Network predictions for parameter variations (LF and LR) in two images. (**a**) Overlay of prediction on the original image. (**b**) Ground truth. (**c**) Predictive mask.

**Figure 8 sensors-24-04924-f008:**
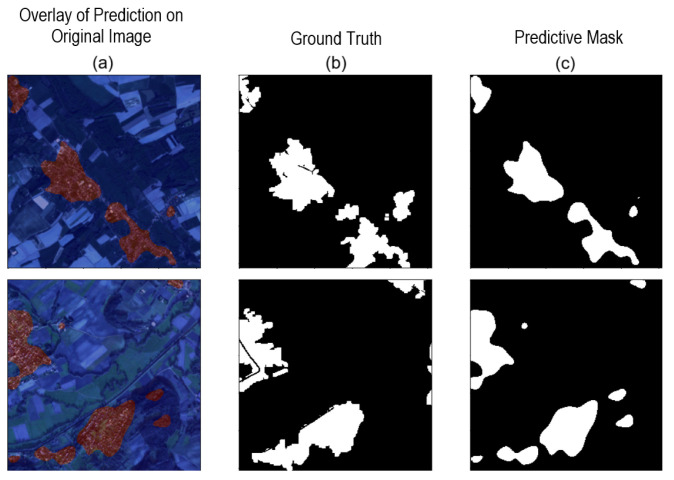
Network predictions after adjusting for training subcategories in two images. (**a**) Overlay of prediction on the original image (**left** panel). (**b**) Ground truth (**central** panel). (**c**) Predictive mask (**right** panel).

**Figure 9 sensors-24-04924-f009:**
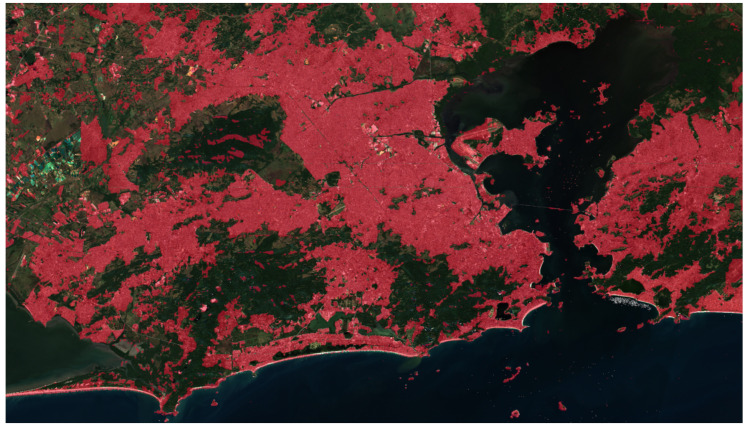
Qualitative results of the network prediction (in red) in the delimited area for the Rio de Janeiro region on 22 July 2023.

**Figure 10 sensors-24-04924-f010:**
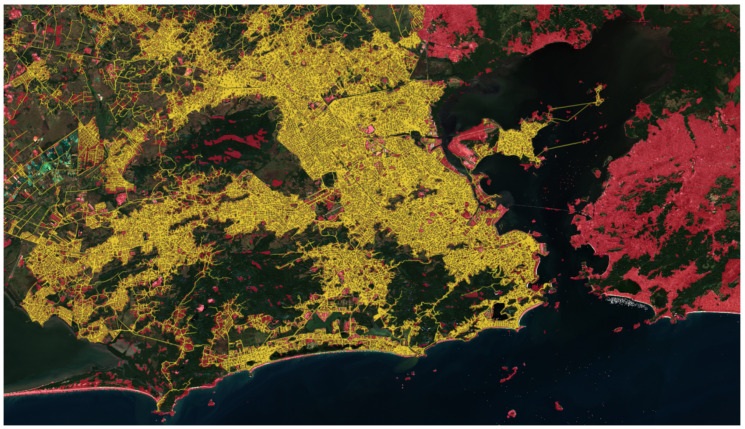
BDGD coverage area (yellow) vs. methodology prediction (red).

**Figure 11 sensors-24-04924-f011:**
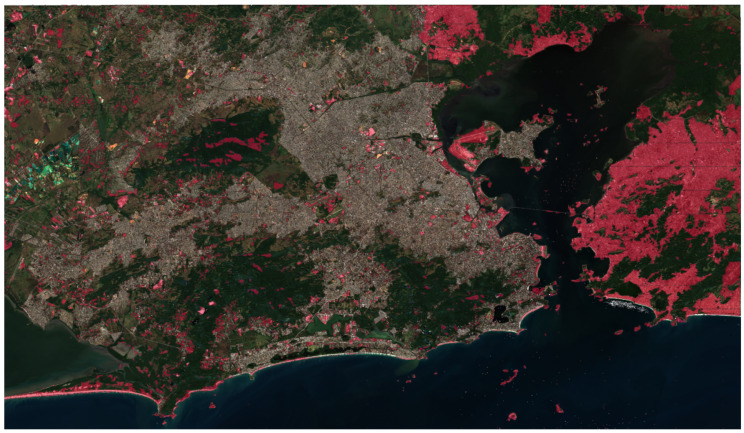
Results of subtracting the BDGD coverage area from the network prediction.

**Figure 12 sensors-24-04924-f012:**
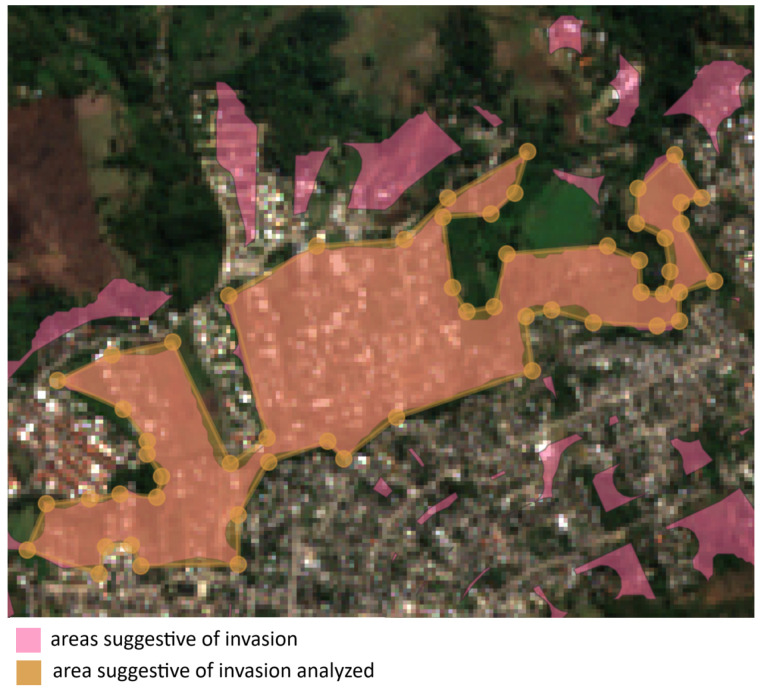
Example 1 of areas suggestive of invasion. The central point (approximate): −22.845202, −43.531593. Average area: 496,922.767 $m^{2}$ or $0.497\;{\mathrm{km}}^{2}$.

**Figure 13 sensors-24-04924-f013:**
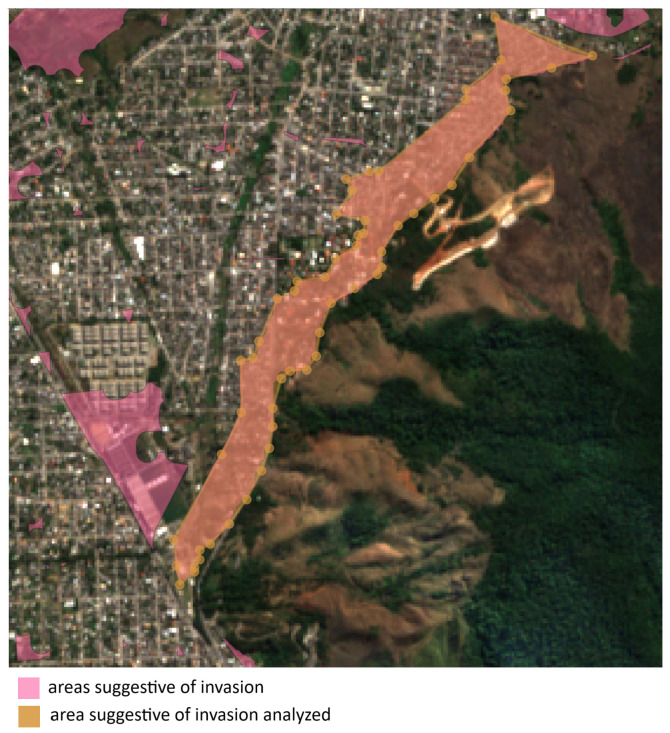
Example 2 of areas suggestive of invasion. The central point (approximate): −22.827001, −43.596853. Average area: 528,803.764 $m^{2}$ or $0.529\;{\mathrm{km}}^{2}$.

**Figure 14 sensors-24-04924-f014:**
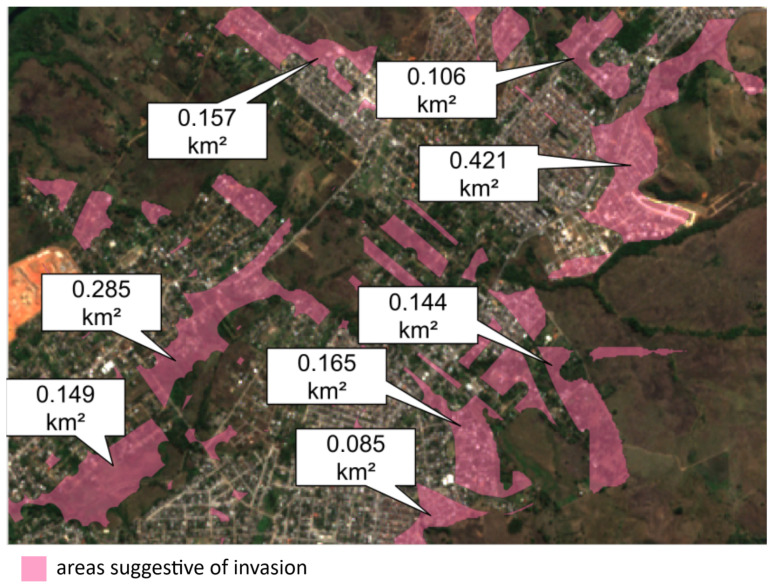
Example 2 of areas suggestive of invasion. Central point (approximate): −22.812473, −43.600857.

**Figure 15 sensors-24-04924-f015:**
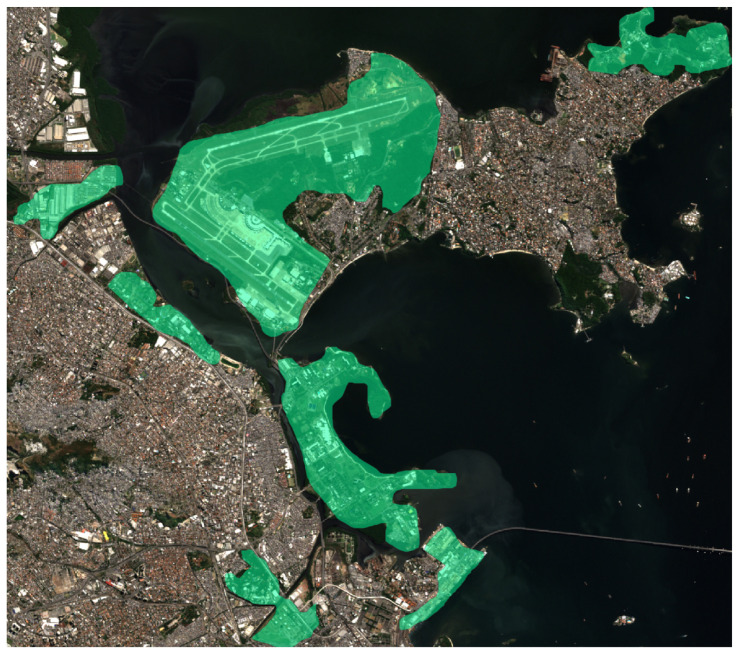
Examples of regions considered false positives.

**Table 1 sensors-24-04924-t001:** Metrics of a network trained with parameter combinations (LF and LR).

Exp.	LF	LR	Acc	F1	IoU	Precision	Recall
1	Dice	$1\times 10^{-4}$	0.9494	0.9257	0.8623	0.9032	0.9494
2	**CE**	$1\times 10^{-4}$	**0.9789**	**0.9591**	**0.9217**	**0.9402**	**0.9789**
3	BCE	$1\times 10^{-4}$	0.9301	0.9522	0.9089	0.9754	0.9301
4	Dice	$1\times 10^{-3}$	0.9221	0.9104	0.8356	0.8996	0.9221
5	CE	$1\times 10^{-3}$	0.9925	0.9580	0.9196	0.9259	0.9925
6	BCE	$1\times 10^{-3}$	0.9165	0.9425	0.8917	0.9703	0.9165
7	CE wgt	$1\times 10^{-3}$	0.8972	0.9359	0.8799	0.9783	0.8972
8	CE wgt	$1\times 10^{-4}$	0.9724	0.9570	0.9178	0.9422	0.9724

The abbreviations CE, BCE, and CE wgt refer to cross-entropy loss, BCE with logit loss, and cross-entropy loss with weights, respectively. Rows 7 and 8: The weights used in cross-entropy loss with weights are 0.7 for the class of interest and 0.1 for the other class, respectively.

**Table 2 sensors-24-04924-t002:** Performance metrics after the reduction of classes using cross-entropy loss function and a learning rate of $1\times 10^{-4}$.

Acc	F1	IoU	Precision	Recall
0.9762	0.9644	0.9313	0.9528	0.9762

**Table 3 sensors-24-04924-t003:** Metrics of the network trained with different types of augmentation.

Augmentation	Acc	F1	IoU	Precision	Recall
Spatial Only	0.9231	0.9476	0.9007	0.9736	0.9231
Pixel Only	0.8836	0.9194	0.8513	0.9584	0.8836
All ^1^	0.9766	0.9521	0.9087	0.9289	0.9766

^1^ Pixel and Spatial augmentation combined.

**Table 4 sensors-24-04924-t004:** Metrics of a network trained with different band combinations.

Bands	Acc	F1	IoU	Precision	Recall
RGB, NIR and SWIR	0.9983	0.9507	0.9063	0.9076	0.9983
All Bands	0.9933	0.9524	0.9093	0.9149	0.9933

**Table 5 sensors-24-04924-t005:** Metrics of the network trained with different types of encoders.

Exp.	Encoder	Acc	F1	IoU	Precision	Recall
1	EFFNet-b0	0.9942	0.9540	0.9113	0.9171	0.9942
2	EFFNet-b1	0.9179	0.9417	0.8901	0.9669	0.91795
3	EFFNet-b2	0.9129	0.9369	0.8817	0.9624	0.9129
4	EFFNet-b3	0.9115	0.9441	0.8944	0.9792	0.9115
5	EFFNet-b4	0.9350	0.9460	0.8979	0.9573	0.9350
6	EFFNet-b5	0.9584	0.9406	0.8883	0.9236	0.9584
7	EFFNet-b6	0.9302	0.9537	0.9117	0.9785	0.9302
8	EFFNet-b7	0.9285	0.9519	0.9085	0.9767	0.9285
9	MVT-0	0.8979	0.9365	0.8808	0.9787	0.8979
10	MVT-1	0.8717	0.9247	0.8603	0.9849	0.8717
11	MVT-2	0.9264	0.9477	0.9007	0.9700	0.9264
12	**MVT-3**	0.9407	0.9588	0.9210	0.9777	0.9407
13	MVT-4	0.9307	0.9551	0.9141	0.9808	0.9307
14	MVT-5	0.8800	0.9291	0.8681	0.9844	0.8800
15	**ResNet34**	0.9572	0.9587	0.9210	0.9603	0.9572
16	ResNet50	0.9436	0.9591	0.9206	0.9752	0.9436
17	ResNet101	0.9160	0.9461	0.8980	0.9784	0.9160
18	ResNet152	0.9217	0.9494	0.9039	0.9789	0.9217

The abbreviation MVT refers to Mix-vision-transform. EFFNet refers to EfficientNet.

**Table 6 sensors-24-04924-t006:** ANOVA results for comparison between loss functions with LR of $1\times 10^{-4}$.

Metric	F Statistic	*p*-Value	95% Confidence Interval
Acc	15.24	0.0048 (adjusted)	[0.93, 0.98]
F1	13.87	0.006 (adjusted)	[0.92, 0.97]
IoU	12.54	0.0084 (adjusted)	[0.86, 0.92]
Precision	11.43	0.0112 (adjusted)	[0.90, 0.95]
Recall	16.32	0.004 (adjusted)	[0.93, 0.98]

**Table 7 sensors-24-04924-t007:** *t*-Test results for comparison between learning rates with CE loss function.

Metric	t Statistic	*p*-Value	95% Confidence Interval
Acc	2.87	0.045	[0.96, 0.99]
F1	2.65	0.052	[0.95, 0.98]
IoU	2.48	0.059	[0.90, 0.93]
Precision	2.71	0.049	[0.93, 0.95]
Recall	2.91	0.043	[0.98, 0.99]

**Table 8 sensors-24-04924-t008:** Metrics of the network in the region of interest of Rio de Janeiro.

Encoder	Acc	F1	IoU	Precision	Recall
ResNet-34	0.9871	0.8764	0.7801	0.7881	0.9871
EFFNet-b6	0.9426	0.9298	0.8688	0.9173	0.9426
**MVT-b3**	0.9269	0.9411	0.8888	0.9557	0.9269

The abbreviations MVT refer to Mix-vision-transform. EFFNet refers to EfficientNet.

## Data Availability

Although the data are public, we have not yet made the data available. Currently, our research team is diligently working to address any technical issues with the relevant organizations so that the data can be made publicly accessible.

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
