# Peer review of "NTL-Unet: A Satellite-Based Approach for Non-Technical Loss Detection in Electricity Distribution Using Sentinel-2 Imagery and Machine Learning"

_sensors, 2024, doi:10.3390/s24154924_

Round 1

Reviewer 1 Report

Comments and Suggestions for Authors

Overview: This paper applied machine learning algorithms for processing satellite images to obtain the mapping of power utility’s urban concession area and the distributor’s geographic data (BDGD) map with the power grid infrastructure. The suspicious areas with a high number of clandestine connections were obtained by crossing these maps.

Comments:

1) Consumer units (CUs) suspected of clandestine connection are those at a certain distance (above 100m) from the power grid. It is a simple strategy for identifying clandestine connections, as it only considers the distance to the power grid. Other variables could have been added to confirm or refute the hypothesis of critical areas. Additionally, that strategy is suitable for extreme cases such as Rio de Janeiro where there are many areas well-known with a high number of clandestine connections in the power grid.

2) The presence of NTL acronym in the paper’s title makes it less self-explanatory. In general, acronyms are not recommended in the title.

3)  Non-technical losses (NTL) acronym is defined several times throughout the Introduction Section. Acronym should only be defined in the first time it appears in the Introduction Section. Henceforth, only the acronym NTL should be used to refer to non-technical losses.

4) The presence of many authors (13 in total) raises suspicions about the equally contribution of all.

5) Lines 26-31: “They can be categorized into three main components: commercial losses, related to measurement failures in regular consumer units; losses due to consumption from inaccessible clandestine connections, caused by unauthorized energy consumption without a formal contract, especially in restricted areas; and other non-technical losses, which are indirectly caused by NTL and are pragmatically considered part of the former [3,4]”. The following excerpt seems obscure and deserves additional explanation: “(..) and other non-technical losses, which are indirectly caused by NTL and are pragmatically considered part of the former”.

6) The introduction lacks specificity and could benefit from a more robust background on non-technical losses in power distribution systems. Some suggestions:

[A] L. Ventura, G. Felix, R. Vargas, L. T. Faria, and J. D. Melo, “Estimation of non-technical loss rates by regions,” Electric Power Systems Research, vol. 223, p. 109685, 2023.

[B] J.B. Leite, J.R.S. Mantovani, “Detecting and locating non-technical losses in modern distribution networks”, IEEE Trans. Smart Grid 9 (2) (2018) 1023–1032,

[C] G.M. Messinis, N.D. Hatziargyriou, “Review of non-technical loss detection methods”, Electr. Power Syst. Res. 158 (2018) 250–266.

[D] M.M. Buzau, J. Tejedor-Aguilera, P. Cruz-Romero, A. Gómez-Expósito, “Hybrid deep neural networks for detection of non-technical losses in electricity smart meters”, IEEE Trans. Power Syst. 35 (2) (2020) 1254–1263.

7) Subsection 2.1.1 about training datasets can be excluded without compromising the understanding of the other sections. This section covers the MultiSenGE database in France regions.

8) Lines 455-458: To facilitate the understanding of the results and investigate potential areas suggestive of invasion, we conducted detailed case studies. In these case studies, a buffer zone of 100 meters around the transmission and distribution lines was selected to represent the potentially covered area. There are clandestine connections within of the buffer zone with a distance of less than 100m to the power grid. Therefore, this method is more suitable for critical regions such as Rio de Janeiro. This comment can be added to the text.

9) Lines 461-462: “However, it’s important to note that depending on the regional context, smaller buffer values may be chosen to adapt to the region’s characteristics. Under what regional conditions the context smaller buffer is chosen?

10)  Figures 14, 15 e 16: insert internal caption for pink and orange colors.

Comments on the Quality of English Language

Minor editing of English language required.

Reviewer 2 Report

Comments and Suggestions for Authors

General comments: This manuscript has proposed and validated the introduced NTL-Unet mainly based on the Sentinel-2 Imagery data for Electricity Distribution over Rio de Janeiro, Brazil, which is quite impressive. The written language of this article is adequate, but the following issues, which aim to enhance the quality, depth, and applicability of the research and ensure that the study contributes meaningful insights into the detection and management of non-technical losses in electricity distribution networks, need special attention.

Comment 1 Overall, it is a good paper, but it needs to be proofread before publication. Can this method be used in other regions, and where is its main potential for application?

Comment 2-Introduction.

1. The method was emphasized, but it seems not comprehensive enough,

2. In addition, the introduction of the research area and data is insufficient

3. From a scientific perspective, what problems or gaps have this article addressed? Therefore, it is worth publishing

Comment 2-Materials and Methods

1. The study utilizes Sentinel-2 satellite imagery with a 10-meter spatial resolution, which is suitable for urban area detection. However, could the authors comment on the impact of this resolution on the detection of smaller built-up areas and whether higher-resolution imagery could improve detection accuracy?

2. The application of the U-Net architecture for semantic segmentation is well-established. Nevertheless, it would be beneficial if the authors could discuss the robustness of their findings against potential biases, such as cloud cover and seasonal variations, which can affect image interpretation.

3. The paper would benefit from a more detailed description of the algorithms and models used for data processing and analysis. This includes the specific parameters chosen for the U-Net architecture and any assumptions made during the process.

4. The study mentions the use of OpenStreetMap (OSM) for pre-annotation, which is subject to user contribution and may not always be up-to-date. Could the authors provide more details on how they validated the OSM data and addressed any inaccuracies in the annotations?

Comment 3-Results.

1. While the study identifies urban areas for NTL detection, a more comprehensive analysis of environmental factors such as vegetation indices, land surface temperature, and urban heat island effects could provide additional insights into the relationship between urbanization and energy distribution.

2. The reliance on remote sensing data necessitates validation with ground-truthing efforts. Could the authors discuss any field validation that was undertaken or plans for such validation to ensure the accuracy of the urban area segmentation?

3. The results require more in-depth analysis. While the study reports promising results, it would be helpful to see a comprehensive summary of all statistical tests conducted, including p-values, confidence intervals, and any corrections for multiple comparisons.

4. Given the multifaceted nature of energy distribution and urban planning, an interdisciplinary approach could be beneficial. The authors might consider collaborating with urban planners, environmental scientists, and energy economists to provide a more comprehensive analysis of the region's vulnerability to NTL. 

Comment 4-Discussions. The discussion should be a dialectical analysis of previous arguments and viewpoints.

1. The study spans from 2000 to 2023, capturing significant urban development. It would be insightful if the authors could discuss whether and how this development may have influenced the observed patterns of NTL and the effectiveness of their detection methodology.

2. The potential implications of the findings for geohazard assessment and management are briefly mentioned. It would be valuable to have a more detailed discussion on how the results could inform early warning systems, risk mitigation strategies, and land-use planning in relation to energy distribution networks.

3. The authors could provide recommendations for future research, such as the potential for using machine learning techniques to predict NTL hotspots or the integration of additional datasets to improve the understanding of urban energy consumption patterns.

Comment 5-Conclusion. The conclusion is too short and lacks detailed data support. 

Comment 6-Languages.

One Writing standard: The hyperparameters mentioned in the text are mentioned in different places, which requires a unified standard. For example, in the method section, the selection criteria and meanings of all parameters mentioned in this article should be clearly defined, and then they can be reasonably cited in the following texts.

Comments on the Quality of English Language

The written language of this article is adequate, but it needs to be proofread before publication.
